# Learning Neuro-symbolic Programs for Language-Guided Robotic Manipulation

**Namasivayam K** [*1]     **Himanshu Singh**[*1]     **Vishal Bindal**[*1]     **Arnav Tuli**[1]

**Vishwajeet Agrawal**[#2]     **Rahul Jain**[#2]     **Parag Singla**[1]     **Rohan Paul**[1]

[1] **Authors are affiliated with IIT Delhi.** [2] **Work done when authors were at IIT Delhi.** [*] **and** [#] **denote equal contribution.**

## Abstract

Given a natural language instruction, and an input and an output scene, our goal is to train a neuro-symbolic model which can output a *manipulation program* that can be executed by the robot on the input scene resulting in the desired output scene. Prior approaches for this task possess one of the following limitations: (i) rely on hand-coded symbols for concepts limiting generalization beyond those seen during training [1] (ii) infer action sequences from instructions but require dense sub-goal supervision [2] or (iii) lack semantics required for deeper object-centric reasoning inherent in interpreting complex instructions [3]. In contrast, our approach is neuro-symbolic and can handle linguistic as well as perceptual variations, is end-to-end differentiable requiring no intermediate supervision, and makes use of symbolic reasoning constructs which operate on a latent neural object-centric representation, allowing for deeper reasoning over the input scene. Our experiments on a simulated environment with a 7-DOF manipulator, consisting of instructions involving reasoning and manipulation over longer time horizons, and scenes richer than those seen during training, demonstrate that our model significantly outperforms existing baselines, particularly in generalization settings.

## 1 Introduction

We address the problem of learning to translate high level language instructions into executable symbolic programs grounded in the robot's state and action space. We focus on multi-step manipulation tasks that involve object interactions such as stacking, and assembling objects referred to by their attributes and spatial relations. We assume the presence of natural supervision from a human teacher in the form of input and output scenes, along with linguistic description of a high-level manipulation task. The goal is to train a task planning model that learns action representations that can be composed to achieve the task. The learning problem is hard since (1) object attributes and actions have to be parsed from the underlying sentence (2) object references need to be grounded given the image and (3) the effect of executing the specified actions has to be deciphered in the image, requiring complex natural language as well as well image level reasoning. Further, the model needs to be trained end-to-end, learning representation for any intermediate sub-goals to be executed to achieve the task.

Prior efforts for this task can be broadly categorized as (i) Traditional approaches which learn a mapping between phrases in the natural language to symbols representing robot state and actions in a pre-annotated dataset[1], [4]–[9]; they lack the flexibility to learn the semantics of concepts and actions on their own, an important aspect required for generalizability (ii) Approaches that model an instruction as a sequence of action labels to be executed, without any deeper semantics, and requiring

36th Conference on Neural Information Processing Systems (NeurIPS 2022).

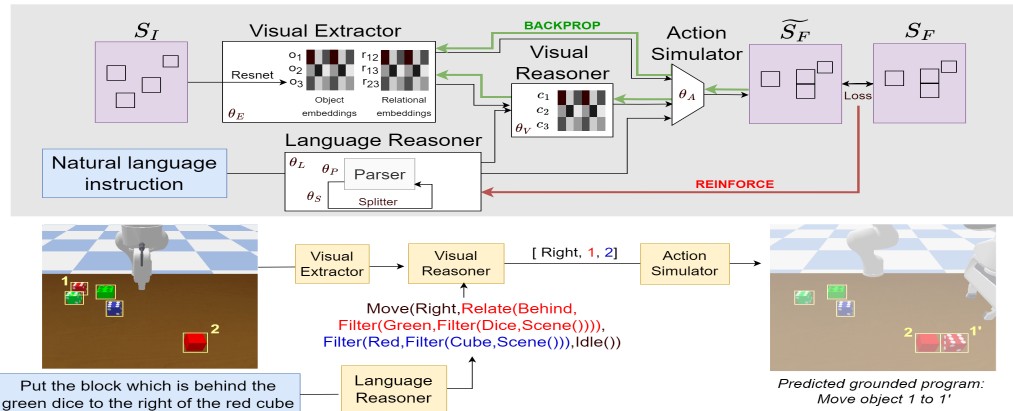

Figure 1: **Model architecture.** The *Visual Extractor* forms dense object representations from the scene image using pre-trained object detector and feature extractor. The *Language Reasoner* auto-regressively induces a symbolic program from the instruction that represents rich symbolic reasoning over spatial and action constructs inherent in the instruction. The *Visual Reasoner* determines which objects are affected by actions in the plan using symbolic and spatial reasoning. The *Action Simulator* determines changes in the world state caused by an action. The model is trained end-to-end without any intermediate sub-goal supervision. The Language Reasoner is trained using REINFORCE and the other modules are optimized using back-prop.

intermediate supervision for sub-goals, which may not be always be available [2], [10]–[16] (iii) Recent approaches which learn the task end-to-end, but have limited reasoning capability both at the level of instruction parsing, and their ability to learn varied action semantics [3], [17]–[23].

In response, we introduce a neuro-symbolic approach for jointly learning grounded concepts that can be composed in manipulation programs that explain how the world scene is likely to affected by the input instruction. Our approach makes use of a Domain Specific Language (DSL) which specifies various concepts whose semantics is learned by the model, in an end-to-end fashion. We build on the concept learning framework by [24], and introduce a model for grounding of concepts in a natural language instruction extracted via a hierarchical parser [25], to those present in the image, and translate them into robot manipulation actions specified as part of an executable programs; a representation that incorporates action composition as well as rich symbolic reasoning. The model outputs a program prescribing a sequence of grounded actions, which when executed by the robot, results in the desired world state. The contributions of this work are: (i) A novel neuro-symbolic model that learns to perform complex object manipulation tasks requiring reasoning over scenes for a natural language instruction, given initial and final world scenes. (ii) A demonstration of how dense representations for robot manipulation actions can be acquired using only the initial and final world states (scenes) as supervision without the need for any intermediate supervision. (iii) Evaluation in instruction, demonstrating robust generalization to novel settings. The data set and code will be publicly released with the final version.

## 2   Problem Formulation

The robot perceives the world state comprising a set of rigid objects placed on a table via a depth sensor that outputs a depth image $S \in \mathbb{R}^{H \times W \times C}$, where $H, W, C$ respectively denote the height, width and the number of channels (including depth) of the imaging sensor. The workspace is co-habited by a human partner who provides language instructions to the robot to perform assembly tasks.The robot's goal is to interpret the human's instruction $\Lambda$ in the context of the initial world state $S_I$ and determine a sequence of low-level motions that result in the final world state $S_F$ conforming to the human's intention. Following [26], [27], planning a complex task is factorized into (i) high-level task planning to determine a sequence of sub-goals, and (ii) the generation of low-level motions to attain each sub-goal. Formally, a semantic model for a manipulation task denoted as $ManipulationProgram(.)$ takes the initial scene $S_I$ and the instruction $\Lambda$ as input and determines a sequence of sub-goals as $(g_0, g_1, \ldots, g_n) = ManipulationProgram(S_I, \Lambda)$. Each sub-goal $g_i$ aggregates the knowledge of the object, $o_i$, to be manipulated and its target Cartesian $SE(3)$ pose $p_i$. This is provided to the low-level motion planner to synthesize the end-effector

trajectory for the robot to execute. Given data $D=\{S_I^i, S_F^i, \Lambda^i\}_{i=1}^M$, the objective is that the final state estimated by simulating the plan inferred by $ManipulationProgram(S_I^i, \Lambda)$ on initial state $S_I^i$, $\tilde{S}_F^i = Simulate(ManipulationProgram(S_I^i, \Lambda; \Theta))$ is close to $S_F^i$. Additionally, we seek strong generalization on novel scenes, instructions and plan lengths beyond those encountered during training, along with interpretability in sub-goals.

## 3    Technical Approach

We propose a neuro-symbolic architecture to solve the task planning problem described in sec 2. Our architecture is inspired by work of Mao et al. [24] for Visual Question Answering (VQA), and our model is trained end-to-end with no intermediate supervision. We assume that the reasoning required to infer the sub-goals can be represented as a program determined by a Domain Specific Language (DSL).The keywords and operators in the DSL, along with the implementation details of the operators is provided in the Appendix A.1. We assume a lexer that identifies all the keywords that are referred to in the instruction $\Lambda$. We do not assume prior knowledge of the semantics of the DSL constructs. Our architecture (ref. Figure 1) consists of the following key modules.

### 3.1    Language Reasoner (LR)

The language reasoner (LR) model deduces a symbolic program that corresponds to the manipulation task implied by the human's utterance to the robot. The symbolic program consists of symbolic reasoning constructs that operate on neural concepts grounded in the state space of objects in the scene and the action space of the robot. Note that representation of a task as a hierarchical and compositional program facilitates deep reasoning. Since a high-level task instruction may imply a sequence of actions, we adopt a hierarchical model where an auto-regressive model first estimates a *split* for the instruction and then estimates a semantic parse for each factored sub-instruction corresponding to a single robot action. Overall the LR module resembles a *seq2tree* architecture that builds on [24], [25]

### 3.2    Visual Extractor (VE)

We assume that we know the gold bounding boxes of the objects in the scene. Following [24], dense objects representations are obtained by passing the bounding boxes and the image through a feature extractor as [28]. For every pair of objects, the relational embedding is obtained from the features of the image that corresponds to the rectangular region containing both the objects. The data association between object proposals is estimated greedily based on cosine similarity between the dense object features in the initial/final scenes.

### 3.3    Visual Reasoner (VR)

The visual reasoner focuses on perfoming spatial object-centric reasoning to resolve the objects that are to be affected by a symbolic program. The visual reasoner takes as input the deduced program from the Language Reasoner and the object features determined by the Visual Extractor and performs *quasi*-symbolic execution of the program on the world state resulting in estimation of objects that are involved in the robot action. In effect, this module resolves the spatial and object reasoning in the program resulting in a plan consisting of a sequence of symbolic actions grounded in the latent object space. Following [24], the visual reasoner consists of neural embeddings and operators corresponding to tokens and operators in the DSL. The output is provided to the Action Simulator (AS) to reason over action consequences on the world state, a task described below.

### 3.4    Action Simulator (AS)

The action simulator is an MLP that learns the semantics of the actions. It takes one-hot representation of the action concept, the initial locations of the object to be manipulated and the reference object respectively and outputs the target location of the former. The outputted target location of the object being manipulated serves as a subgoal for the low-level motion planner. In our experiments, the location $loc \in \mathbb{R}^5$ of an object is determined by the corners b = $(x_1, y_1, x_2, y_2)$ of the bounding box and the depth $d$ of the object from the camera face.

Overall, the model can be summarized as follows:

- $P \leftarrow LR(\Lambda; \theta_P, \theta_S)$, where P is a symbolic program composed of DSL constructs.
- $\Pi \leftarrow VR(P, VE(S_I; \theta_E); \theta_V)$. The visual reasoner then grounds P and outputs $\Pi$, the grounded program. For example, in Figure 1, red and blue subprograms are grounded to object 1 and 2 respectively.
- $G \leftarrow AS(\Pi; \theta_A)$. The action simulator then takes $\Pi$ and returns a sequence of sub-goals, $G = (g_0, g_1, ..., g_{n-1})$, for the motion planner.

where, $\theta$ represents the corresponding parameters for each module.

## 3.5  Loss Function and Model Training

Given a single-step instruction $\Lambda$, the parser predicts a program, P, which is grounded and executed on the initial scene, $S_I$ to get the predicted final locations of the objects $\{\widetilde{loc}_F^i\}_{i=1}^N$. Let $\{loc_F^i\}_{i=1}^N$ be the true locations in the gold final state, $S_F$. As mentioned above $loc = (b, d)$, where $b$ is the corners of bounding box and $d$ is the depth. The loss function $L_{act} := \alpha \sum_i^N \|\widetilde{loc}_F^i - loc_F^i\| + \beta(1 - \text{IoU}(\widetilde{b}_F^i, b_F^i))$ is used to train the action simulator and the visual modules. Since, there is no explicit supervision to the parser, we train the parser using the reinforcement learning policy gradient algorithm REINFORCE with the reward set to $-L_{act}$. An explicit expectation (subtracting the mean action loss as the baseline) is computed over all programs to inform the loss, ameliorating the noisy rewards from the action simulator. In essence, the Language Reasoner is trained using REINFORCE and the other modules parameters are optimized using back prop. (see Fig 1) The details of the curriculum used in the training is provided in the Appendix A.2.

## 4  Experiments

In our experiments we study (i) whether our method can infer programs to translate instructions to desired goal states, (ii) the extent of generalization to novel instructions and world states and (iii) the model's ability to generalize to multiple-step plans having been trained on simpler plans. A demonstration of the learnt model using a simulated Franka Emika is given in the Appendix A.5

Table 1: Accuracy Comparison for the Proposed Model and the Baseline. (BB: Bounding Boxes)

| Model | Overall | | | Single-step | | Double step | | Simple | | Complex | |
|---|---|---|---|---|---|---|---|---|---|---|---|
| | IOU | IOU-M | Program (Action/Subj/Pred) | IOU | IOU-M | IOU | IOU-M | IOU | IOU-M | IOU | IOU-M |
| Baseline | 0.77 | 0.55 | –/0.81/0.76 | 0.80 | 0.56 | 0.71 | 0.52 | 0.90 | 0.71 | 0.64 | 0.31 |
| Ours | 0.87 | 0.72 | 0.99/0.99/0.94 | 0.91 | 0.64 | 0.87 | 0.62 | 0.92 | 0.73 | 0.87 | 0.64 |

## 4.1  Experimental Setup

**Data collection.** The datset is collected using a PyBullet table top environment using a simulated Franka Emika Panda robot arm. 6250 synthetic scenes are sampled with $3 - 5$ blocks (with variations in color and type attribute) along with a natural language instruction for each scene. Each data point consists of the initial scene, final scene and a language instruction without any sub-goal supervision. The main corpus consists of both single-step and double-step commands, along with sentences of different complexities:- *simple* and *complex*. Complex sentences involve reasoning on inter-object relationships, while simple sentences reason over individual object features only. We train the model only on this corpus. However, we generate two additional test datasets of size 1000 to evaluate the generalization ability of our model. The first has richer scenes ($4 - 10$ objects in each scene) and the second has instructions with longer action sequences (multi-step instructions ranging from $3 - 7$ steps) than the examples in the main corpus.

**Neural-only baseline.** Note that, most of the recent works like [2] uses sub-goal supervision, and hence, we cannot use them as baseline to ensure fairness in the evaluation. To study the effectiveness of the approach we construct a purely neural baseline inspired and adapted from [29]. We provide an object-centric world representation to the baseline without assuming sub-goal supervision for

comparison with our setting. Moreover, our model and the baseline share the instruction encoder, the action simulator and the visual extractor. Additionally, baseline includes an action decoder that gives a dense representation for the action at each timestep, and attention networks that give two probability distributions over the objects. These distributions are then used to get the location of the arguments (subject and predicate), as a weighted linear combination of the locations of the objects in the scene. The action representation, and the two argument locations are passed to the action simulator to get the predicted location of the subject. Here, *predicted location* includes both the bounding box and depth of the moved object. The entire architecture is neural and is trained end-to-end via back propagation.

**Metrics.** The following metrics are used for evaluation: (i) *Intersection over Union (IOU)*: of the predicted bounding boxes in the final scene in comparison with the ground truth bounding boxes extracted from the original demonstration. The IOU metric is calculated in 2D in the image space (assuming a static camera viewing the scene). Average IOU over all objects in the scene and mean IOU for objects moved during execution, termed IOU-M is reported. (ii) *Program Accuracy*: The grounded program inferred for an (instruction, scene)-pair using the proposed model is compared with the ground truth program (manually annotated). We separately report the grounding accuracy for the subject and predicate of our action (assumed binary) and the accuracy of the predicted action inferred from the instruction. Since there is no explicit notion of grounded actions in the baseline, we do not report this metric for the baseline.

## 4.2 Results

**(i) Performance:** we use a 80 : 20 train:test split of the main corpus for accuracy comparison. Table 1 reports the performance of our model and the baseline on the test set. Our model outperforms the baseline overall. For instructions with complex reasoning (resolution of binary spatial relations) involved, the proposed model outperforms the baseline by 33 points in the IOU-M metric. We attribute this to the disentangled representations of visual and action concepts that allow efficient complex reasoning and manipulation.

**(ii) Generalization to richer environments:** The proposed model was first trained on scenes having up to 5 objects only, and then tested on scenes having up to 10 objects. Even on larger scenes, the model is able to interpret the correct object and move it to the correct position with marginal decrease in accuracy. The improved generalization demonstrated by the model can be attributed to reliance on an object-centric world model and the ability to learn dense disentangled representations for spatial and action concepts.

**(iii) Generalization to longer plans:** We evaluate model generalization to inferring plans extending to time horizons beyond those observed during training. The model is first trained on instructions conveying plans with up to few (1-2 step) manipulation actions and evaluated on instructions with longer action sequences. The model is able to perform scene manipulation up to 7 steps without any appreciable drop in accuracy (See Figure 2b). We attribute this to the modular structure of our approach compared to the baseline.

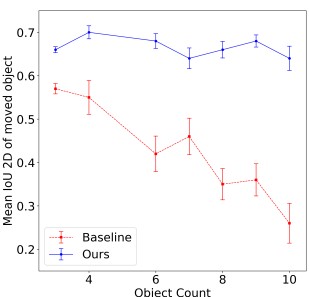

(a) IoU vs # of objects

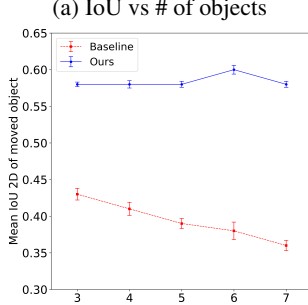

(b) IoU vs varying # of steps

Figure 2: Performance in Generalization Settings

## 5 Conclusions

We present a neuro-symbolic architecture that learns grounded manipulation programs via visual-linguistic reasoning for instruction understanding over a given scene, to achieve a desired goal. Unlike previous work, we do not assume any sub-goal supervision, and demonstrate how our model can be trained end-to-end. Our experiments show strong generalization to novel scenes and instructions compared to a neural-only baseline. Limitations include sparse action space, tokenized language and lack of uncertainty modeling in action semantics. Directions for future work include dealing with richer instruction space including looping constructs, real-time recovery from errors caused by faulty execution, and working with real workspace data.

## Acknowledgements

We thank anonymous reviewers for their insightful comments that helped in further improving our paper. Parag Singla was supported by the DARPA Explainable Artificial Intelligence (XAI) Program with number N66001-17-2-4032, IBM AI Horizon Networks (AIHN) grant and IBM SUR awards. Rohan Paul acknowledges Pankaj Gupta Young Faculty Fellowship for support. Any opinions, findings, conclusions or recommendations expressed in this paper are those of the authors and do not necessarily reflect the views or official policies, either expressed or implied, of the funding agencies.

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

# A  Appendix

## A.1  Domain Specific Language (DSL)

Table 2 list all the keywords and the operators that we have in our DSL.

| Keywords and their classes | |
|---|---|
| **Object-level concepts** | **Other concepts** |
| Color: {red, blue, cyan,...} | **Relation**: {left, behind, front,...} |
| Type: {Cube, Lego, Dice} | **Action**: {MovRight, MovTop,...} |
| **Operators: ( Input → output)** | |
| Scene : None → ObjSet | Unique: ObjSet → Obj |
| Filter : (ObjSet, ObjCpt)→ ObjSet | Relate : (Obj, RelCpt) → ObjSet |
| Move : (ActCpt, World) → World | Idle : World → World |
| ObjectSet $\in \mathbb{R}^N$, N = Num objects, Object = one-hot ObjectSet | |
| World = $\{(b_i, d)\}_{i=1}^N \in \mathbb{R}^5$, bounding boxes and depth for all objects | |

Table 2: Domain Specific Language.

## A.2  Curriculum Strategy

Since there are multiple stages in our pipeline, and the supervision is available only at the end, it is important to define a curriculum in order to train effectively. In particular, we need to first train in simpler settings, followed by freezing of certain modules, and then move on to more complex instructions. Such curriculum training has been found to be effective in prior neuro-symbolic approaches [24] as well. Our curriculum consists of the following steps: (I) We first train on single step commands, and with only selection on a single attribute type (e.g., color or size) for any given object. This allows our visual reasoner to learn concept embeddings and attribute neural operators, and the action simulator to learn disentangled action representations. (II) In the second step, we allow for instructions with selection on multiple attribute types. In this step, the action simulator is kept frozen; this can be done since action simulator does not directly depend on the linguistic variations or the number of attribute types being used to qualify the objects. (III) In the last step, we allow for instructions involving multiple steps. In this step of curriculum training, we freeze the rest of the pipeline, and only train the splitter (see appendix A.3) in the Language Reasoner which splits a given instruction into multiple single step instructions. This can be done, since rest of the pipeline can operate as earlier, once a multi-step instruction has been split into its respective single-step equivalents. The entire model is then fine-tuned jointly after curriculum training.

## A.3  Training the splitter

Once we have trained the other modules on single step commands, we train the splitter on all one and two step commands in the training set. The splitter computes the probability $\mathcal{S}(s; \theta_s)$ of breaking the sentence on each token $s$ of the first $L_{max}$ tokens. The training objective, for a sentence $\Lambda$, can be described as

$$\theta_S \leftarrow \text{argmin}\Big(\mathbb{E}_{s \sim \mathcal{S}}\big[L_{act}(Compose(Parser(L_{0:s}), Parser(L_{s+1:})))\big]\Big)$$

where $Compose$ takes input symbolic programs generated by the parser for single step instructions and composes them hierarchically. Since all other modules are already trained for single step sentences. The splitter learns to split the large sentences at the correct positions.

## A.4  Scene reconstruction

We additionally train a neural model to synthesize the scene corresponding to each sub-goal, given only the initial scene and predicted object locations. This enables us to visualise the scene modification without the need of execution by a robot manipulator along with providing interpretability to the model's latent program space. The reconstruction architecture is adapted from [30], where the scene graph is constructed with nodes having object features and bounding boxes. This graph is updated with predicted bounding boxes at each step, yielding generated scenes. Presence of initial and final scenes in our data means we can train in a supervised manner, unlike [30].

## A.5 Demonstration on a Simulated Robot

We demonstrate the learned model for interpreting instructions provided to a simulated 7-DOF Franka Emika manipulator in a table top setting. The robot is provided language instructions and uses the model to predict a program that once executed transitions the world state to the intended one. The 2-D bounding boxes predicted by the action simulator are translated to 3-D coordinates in the world space via a learnt MLP using simulated data. The predicted positions are provided to a low-level motion planner for trajectory generation with crane grasping for picking/placing. Each step of the robot simulation is then performed by grasping the object at the initial location, moving the gripper to the final predicted location, and releasing the gripped object. Figure 3 shows execution by the robot manipulator on complex instructions, scenes having multiple objects, double step relational instructions, and multi-step instructions. We also visualise reconstruction of the moved objects before each step of the actual execution. The structural similarity index (SSIM) for the reconstruction model is 0.935.

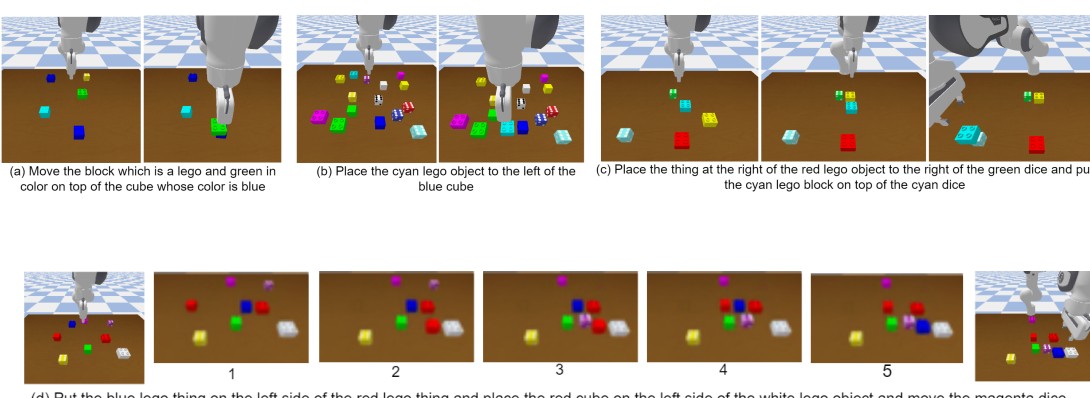

(a) Move the block which is a lego and green in color on top of the cube whose color is blue

(b) Place the cyan lego object to the left of the blue cube

(c) Place the thing at the right of the red lego object to the right of the green dice and put the cyan lego block on top of the cyan dice

(d) Put the blue lego thing on the left side of the red lego thing and place the red cube on the left side of the white lego object and move the magenta dice the right of the green box and move the red box on the left side of the blue lego thing and put the blue lego object to the left of the white lego thing

Figure 3: Execution of robot manipulator on (a) compound instructions, (b) scene with 15 objects, (c) double step instruction with relational attributes, (d) 5-step instruction. (d) also shows reconstruction of the predicted scene before each step of the simulation

