# OpenReview forum: "Learning Neuro-symbolic Programs for Language-Guided Robotic Manipulation"
_NeurIPS.cc/2022/Workshop/nCSI — nCSI WS @ NeurIPS 2022 Poster_

### Official Review · Reviewer_Sxds · 2022-10-08
**Well-written paper with a good technical quality**

**Rating:** 2
**Confidence:** 2

**Review:**

## Summary
The paper proposes a new neuro-symbolic framework, which can perform learning to translate instructions to grounded robot plans. The addressed problem is to solve planning problems where the input is given as a pair of a state as a depth image and natural language instruction. The task is difficult since the agent needs to understand both the visual state and the natural language instruction and to perform planning on top of them. To address the problem, the paper proposes a new framework, that consists of language reasoner, visual extractor, visual reasoner, and action simulator. Each component processes information so that the entire system can execute symbolic programs defined in the DSL in a differentiable manner to solve the visual planning problem. In the experiment, the proposed approach outperformed a neural baseline, and moreover, it showed strong generalization results in terms of the number of objects and the number of steps of the planning.

## Pros
- The paper is very well written
- The paper has a good technical quality, i.e., everything is formulated without technical flaws
- The proposed approach is novel in the sense that it solves visual planning problems using the NS-CL approach beyond VQA tasks
- The paper leads to many important applications

## Cons
The paper is overall well-written, however, I noticed some minor points that can be addressed.
- In Fig. 1, Visual Extractor has Relational embeddings, however, it is not explained in the main text. If not the case I'm missing something, please explain it properly somewhere in the paper.
- In line 107, I think Action simulator should be capitalized as Action **S**imulator

### Questions
- The limitation of the proposed approach is not discussed in the paper. What would be the limitation?
- Is the proposed framework capable of parallelized batch computation? The implementation of neural networks in general can process a given batch of examples in parallel on GPUs. For the proposed approach, if the user gives several examples as input, are they processed in parallel? If not, how long does it take to train the proposed model? Would it take longer compared to the neural-based baseline?

---

### Official Review · Reviewer_ebFe · 2022-10-14
**Interesting**

**Rating:** 2
**Confidence:** 2

**Review:**

This work is concerned with developing a model that can produce a manipulation program in order for a robot to manipulate its surrounding to reach a goal state given an initial state. The work focuses here on a neruo-symbolic approach that includes several specialized submodules. Despite these differences all submodules can be trained end-2-end without intermediate supervision. The experimental results are promising.
Overall this is a well written paper with a very relevant topic and proposed method for this workshop. I am hesitant on giving this paper the highest score due to two reasons that are somewhat correlated. The overall methods section in "Technical Approach" is quite minimilistic and scarce in details. E.g. particularly the description of the visual extractor is kept very short. Even if it is mainly based on previous work I believe it would be helpful to have some more information overall in this technical approach section. This leads to one of the more important contributions of the work, the single loss function (section 3.5), to be less obvious, i.e. how the gradient flows through to all initial modules. And particularly in my view this is the most interesting feat of the work, i.e. that training all submodules works via this one loss. Maybe adding more explicit mathematical notations in Section 3 could help for the comprehensability here.

---

### Meta-Review · Area_Chair_769K · 2022-10-19

**Recommendation:** 2
**Confidence:** 3

**Metareview:**

As both reviewers stated, this paper is a good fit for the workshop, with several positives.
I am not repeating all of them here, since I agree with the reviewers' assessment comments.
From the a given a natural language instruction and an input and an output scene,
the paper investigates how to  train a neuro-symbolic model which  manipulates
a program that can be executed by a robot on the input scenes and generates a goal state.
The authors carry out experiments to demonstrate how the neuro-symbolic model
is end-to-end and show generalization to novel scenes and instructions.
I recommend acceptance of this paper, as it can lead to relevant directions in neuro-symbolic robotics.

---

### Decision · Program_Chairs · 2022-10-20

Accept (Poster)